# Keeping Cell Death Alive: An Introduction into the French Cell Death Research Network

**DOI:** 10.3390/biom12070901

**Published:** 2022-06-28

**Authors:** Gabriel Ichim, Benjamin Gibert, Sahil Adriouch, Catherine Brenner, Nathalie Davoust, Solange Desagher, David Devos, Svetlana Dokudovskaya, Laurence Dubrez, Jérôme Estaquier, Germain Gillet, Isabelle Guénal, Philippe P. Juin, Guido Kroemer, Patrick Legembre, Romain Levayer, Stéphen Manon, Patrick Mehlen, Olivier Meurette, Olivier Micheau, Bernard Mignotte, Florence Nguyen-Khac, Nikolay Popgeorgiev, Jean-Luc Poyet, Muriel Priault, Jean-Ehrland Ricci, Franck B. Riquet, Santos A. Susin, Magali Suzanne, Pierre Vacher, Ludivine Walter, Bertrand Mollereau

**Affiliations:** 1Cancer Cell Death Laboratory, Part of LabEx DEVweCAN, Cancer Initiation and Tumoral Cell Identity Department, CRCL, 69008 Lyon, France; 2Apoptosis, Cancer and Development Laboratory—Equipe Labellisée ‘La Ligue’, LabEx DEVweCAN, Centre de Recherche en Cancérologie de Lyon, INSERM U1052-CNRS UMR5286, Claude Bernard University Lyon 1, 69008 Lyon, France; benjamin.gibert@lyon.unicancer.fr (B.G.); patrick.mehlen@lyon.unicancer.fr (P.M.); olivier.meurette@lyon.unicancer.fr (O.M.); 3INSERM U1234, Pathophysiology, Autoimmunity, and ImmunoTHERapies (PANTHER), Normandie University, UNIROUEN, 76000 Rouen, France; sahil.adriouch@univ-rouen.fr; 4CNRS, Institut Gustave Roussy, Aspects Métaboliques et Systémiques de L’oncogénèse pour de Nouvelles Approches Thérapeutiques, Université Paris-Saclay, 94805 Villejuif, France; catherine.brenner@universite-paris-saclay.fr (C.B.); sletlana.dokudovskaya@universite-paris-saclay.fr (S.D.); 5Laboratory of Biology and Modelling of the Cell, Ecole Normale Superieure of Lyon, CNRS, UMR 5239, INSERM U1293, University Claude Bernard Lyon 1, 46 allee d’Italie, F-69364 Lyon, France; nathalie.davoust-nataf@ens-lyon.fr (N.D.); ludivine.walter@ens-lyon.fr (L.W.); 6Institut de Génétique Moléculaire de Montpellier, Université de Montpellier, CNRS, 34293 Montpellier, France; solange.desagher@igmm.cnrs.fr; 7INSERM UMR-S-U1172, Lille Neurosciences & Cognition, University of Lille, CHU Lille, F-59000 Lille, France; david.devos@chru-lille.fr; 8Institut National de la Santé et de la Recherche Médicale (INSERM) INSERM LNC UMR1231, LabEx LipSTIC, 21000 Dijon, France; laurence.dubrez@u-bourgogne.fr; 9INSERM U1124, Université Paris Cité, F-75006 Paris, France; jerome.estaquier@crchudequebec.ulaval.ca; 10Centre de Recherche en Cancérologie de Lyon, U1052 INSERM, UMR CNRS 5286, Centre Léon Bérard, Université Lyon I, 28 Rue Laennec, 69008 Lyon, France; germain.gillet@lyon.unicancer.fr (G.G.); nikolay.popgeorgiev@univ-lyon1.fr (N.P.); 11Hospices Civils de Lyon, Centre de Biologie Sud, Centre Hospitalier Lyon Sud, Chemin du Grand Revoyet, 69495 Pierre Bénite, France; 12LGBC, UVSQ, Université Paris-Saclay, 78000 Versailles, France; isabelle.guenal@uvsq.fr (I.G.); bernard.mignotte@uvsq.fr (B.M.); 13EPHE, Université PSL, 75014 Paris, France; 14INSERM CRCI2 NA, Université de Nantes, F-44000 Nantes, France; philippe.juin@univ-nantes.fr; 15Institut Universitaire de France, Centre de Recherche des Cordeliers, Equipe Labellisée par la Ligue Contre le Cancer, Université de Paris, Sorbonne Université, INSERM U1138, 75006 Paris, France; kroemer@orange.fr; 16Contrôle de la Réponse Immune B et Lymphoproliférations, CRIBL, Université Limoges, UMR CNRS 7276, INSERM 1262, 87025 Limoges, France; patrick.legembre@inserm.fr; 17Department of Developmental and Stem Cell Biology, Institut Pasteur, Université de Paris Cité, CNRS UMR3738, F-75015 Paris, France; romain.levayer@pasteur.fr; 18UMR 5095, CNRS, Université de Bordeaux, 33000 Bordeaux, France; manon@ibgc.cnrs.fr; 19INSERM, Lipides Nutrition Cancer, UMR 1231, Université de Bourgogne Franche-Comté, 21000 Dijon, France; omicheau@u-bourgogne.fr; 20Cell Death and Drug Resistance in Hematological Disorders Team, Centre de Recherche des Cordeliers, UMRS 1138, INSERM, Sorbonne Université, F-75006 Paris, France; florence.nguyen-khac@psl.aphp.fr (F.N.-K.); santos.susin@sorbonne-universite.fr (S.A.S.); 21Groupe Hospitalier Pitié-Salpêtrière, Assistance Publique—Hôpitaux de Paris, F-75013 Paris, France; 22Institut Universitaire de France (IUF), F-75013 Paris, France; 23INSERM, HIPI, Université Paris Cité, F-75475 Paris, France; jean-luc.poyet@inserm.fr; 24CNRS, IBGC, UMR 5095, University Bordeaux, F-33000 Bordeaux, France; muriel.priault@ibgc.cnrs.fr; 25INSERM C3M, Equipe Labellisée LIGUE Contre le Cancer, Université Côte d’Azur, 06200 Nice, France; jean-ehrland.ricci@unice.fr; 26Cell Death and Inflammation Unit, VIB-UGent Center for Inflammation Research (IRC) & Department of Biomedical Molecular Biology (DBMB), Ghent University, B-9052 Ghent, Belgium; franck.riquet@irc.vib-ugent.be; 27UMR 8523, Physique des Lasers Atomes et Molécules, Université de Lille, CNRS, 59000 Lille, France; 28Centre de Biologie Intégrative, CNRS/UMR 5088, Université Toulouse III, 31062 Toulouse, France; magali.suzanne@univ-tlse3.fr; 29INSERM U1218, Institut Bergonié, 33000 Bordeaux, France; pierre.vacher@inserm.fr

**Keywords:** cell death, apoptosis, necrosis, cancer

## Abstract

Since the Nobel Prize award more than twenty years ago for discovering the core apoptotic pathway in *C. elegans*, apoptosis and various other forms of regulated cell death have been thoroughly characterized by researchers around the world. Although many aspects of regulated cell death still remain to be elucidated in specific cell subtypes and disease conditions, many predicted that research into cell death was inexorably reaching a plateau. However, this was not the case since the last decade saw a multitude of cell death modalities being described, while harnessing their therapeutic potential reached clinical use in certain cases. In line with keeping research into cell death alive, francophone researchers from several institutions in France and Belgium established the French Cell Death Research Network (FCDRN). The research conducted by FCDRN is at the leading edge of emerging topics such as non-apoptotic functions of apoptotic effectors, paracrine effects of cell death, novel canonical and non-canonical mechanisms to induce apoptosis in cell death-resistant cancer cells or regulated forms of necrosis and the associated immunogenic response. Collectively, these various lines of research all emerged from the study of apoptosis and in the next few years will increase the mechanistic knowledge into regulated cell death and how to harness it for therapy.

## 1. General Introduction

Programmed or regulated cell death is an essential process, which ensures cellular homeostasis of living organisms [1]. Among the different forms of regulated cell death, apoptosis is undoubtedly the best studied. Within the last three decades a tremendous number of studies have reported not only the identification and characterization of apoptotic factors but also their role in physiopathology. Initiated by either a cell extrinsic or intrinsic death stimulus, apoptosis ultimately converges to mitochondrial outer-membrane permeabilization (MOMP) and caspase cysteine protease activation. MOMP and subsequent caspase activation are regulated by anti-apoptotic (e.g., BCLxL, BCL-2, MCL1) and pro-apoptotic BCL2 proteins (such as BAX, BAK, BID, BIM or PUMA) and inhibitors of apoptosis proteins (IAPs) [2]. Although some could have predicted a plateau in the number of studies on apoptosis and other forms of regulated cell death, the number of publications has linearly increased every year with nearly 40,000 publications in 2020 (National Library of Science). This rising interest is due to multiple parameters, which include continuous efforts in the characterization of the core apoptosis machinery in animal development and pathologies, but also the appearance of emerging fields linked to cell death, including studies on alternative function of apoptotic factors, alternative forms of regulated cell death and their involvement in various pathologies.

The French Cell Death Research Network (FCDRN) is actively taking part in further understanding apoptosis in development, physiopathology and in the emergence of new research fields related to cell death. During the last few years, the FCDRN has organized several meetings between its members to foster discussion and collaborations. During these meetings, FCDRN researchers felt the need to spread the idea that the cell death field has dramatically evolved, expanded and is still vivid. In this review, we compile a non-exhaustive list of 25 contributions on cell death research from both France and Belgium (Figure 1). It is an attempt to reflect the current state on cell death research in the FCDRN community, and can be summarized as follows:

Years after the identification of the core apoptotic pathway in *Caenorhabditis elegans*, current scientific challenges on developmental apoptosis focus on the understanding of the dynamics of apoptosis in animal development, i.e., how dying cells interact with their cellular environment in developing tissues. This recent research is taking advantage of technological breakthroughs in live imaging, different novel fluorescence reporters and cell tracing combined with interdisciplinary approaches to tackle the fascinating questions of mutual relationships between mechanical forces and apoptosis during *Drosophila* development (**Suzanne** and **Levayer** teams).

The core apoptotic signaling network has been studied in great detail in various cellular and animal models [3,4,5,6]. However, there is still much to learn on apoptosis regulation, depending on the cellular contexts and tissues. To study these fine regulations, researchers undertake evolutionary approaches on Bcl2 family protein members and mitochondrial apoptosis in *Saccharomyces cerevisiae* (**Manon** team) and *Drosophila melanogaster* (**Guenal/Mignotte** team). Another strategy undertaken is the study of the regulation of Bcl2 family proteins by the TRIM proteins of the ubiquitin proteasome machinery (**Desagher** team). The TNFR family members-dependent regulation is also under deep scrutiny, by investigating apoptotic and non-apoptotic roles of TRAIL receptors in cancer cells (**Micheau** team).

The recent understanding that apoptotic factors have non-canonical roles in various cellular processes [7,8,9,10,11] raised considerable interest in the international scientific community and researchers of the FCDRN. Non-canonical functions of apoptotic proteins, such as cIAPs, Bcl2 family members and caspases have been shown to contribute to cellular differentiation, morphogenesis or the cross-talks between apoptosis and autophagy pathways. For example, cIAP1 and cIAP2 regulate cell differentiation and proliferation in hematopoietic stem cells (**Dubrez** team). Bcl2 family members exhibit non-canonical functions during embryonic vertebrate development (**Gillet** team). A non-apoptotic function of the Bcl2 family member, Bcl-xL, is also central for the regulation of autophagy (**Priault** team). A cross-talk between apoptosis and autophagy pathways is also investigated in cancer cells, in which cell death is induced by Kremen-1 dependence receptor (**Meurette** team). Unexpectedly, non-canonical functions of apoptotic proteins can also promote oncogenesis. This is the case for caspases, the executioner of apoptosis, which at low levels exhibit functions in cancer cell motility (**Ichim** team). Furthermore, oncogenesis can also be promoted by a non-canonical function of secreted CD95L (**Legembre** team).

The dogma that apoptosis is the only form of regulated cell death involved in animal development and pathologies was challenged by the understanding that necrotic cell death is not only an accidental process but can be genetically controlled. This led to the identification and characterization of different regulated necrosis pathways including necroptosis, ferroptosis, pyroptosis and parthanatos [12,13]. It is now generally accepted that regulated necrosis is also involved in the control of cell homeostasis during differentiation, such as for p53-dependent elimination of excess germ cells during spermatogenesis in both vertebrates and invertebrates (**Mollereau** team). An important focus is also given to the study of regulated necrosis in pathologies, which has led to translational research in neurodegenerative diseases and cancer. For example, ferroptosis has an emerging and growing role in neurodegenerative diseases such as Parkinson disease and amyotrophic lateral sclerosis (**Devos** team).

In cancer therapy, an important challenge is to determine if the elimination of resistant cancer cells to apoptosis can be achieved by activating caspase independent-cell death. This is in particular studied in relation with the energy metabolism, which regulates cancer cell sensitivity to chemotherapeutic treatments (**Ricci** team). Moreover, inducing caspase-independent cell death by activating the CD47 receptor is a promising strategy to overcome resistance to treatment in chronic lymphocytic leukemia (CLL) (**Susin/Nguyen-Khac** team). Interestingly, the study of resistance to treatment can be also achieved by investigating the acquired resistance to apoptosis or by activating non-canonical apoptotic pathways in cancer cells. Hence, the study of mitochondrial apoptosis can still reveal important insights on the acquired resistance to apoptosis in cancer cells (**Juin** team). Another important factor that confers cancer cell resistance to apoptosis is AAC-11. One effective way to overcome apoptosis resistance is to develop Leuzine-Zipper Derived Peptides (LZDP) interacting with and inhibiting AAC-11 (**Poyet** team).

An alternative strategy employed by cancer cells is to overcome death receptor-induced apoptosis. This is the case for the dependence receptor pathway, non-canonical pro-apoptotic receptors which led to important breakthroughs in cancer therapy [14]. The discovery that inhibiting dependence receptors by excess of their respective ligands contributes to apoptosis resistance in several cancer models led to the development of blocking antibodies used now in early-phase clinical trials (**Mehlen** team). Further exploitation of dependence receptor stimulation for cancer cell killing is undertaken by targeting Netrin-3, an understudied member of the Netrin ligand family in cancer, whose expression correlates with cancer aggressiveness in neuroblastoma and small cell lung cancer (SCLC) (**Gibert** team).

The study of the anti-cancer immunogenicity of different forms of cell death raised considerable interest in the scientific community [15]. First, the comprehension that apoptotic cells, which were at first thought to be non-immunogenic, release in fact danger-associated molecular patterns (DAMPs), was an important turning point opening novel perspectives in stimulating the innate and cognate immune response to fight cancer (**Kroemer** team). Second, the characterization of the pro-inflammatory response elicited by necroptotic cells has led to important studies in characterizing the associated signaling dynamics of pathways, such as ERK1/2, involved in this anti-tumoral response (**Riquet** team). Finally, limiting the side-effects of anti-cancer therapy is undertaken by identifying molecules stimulating survival pathways to protect normal cells (**Brenner** team) or new therapeutic strategies (**Adriouch** team).

In the context of host-pathogen interactions, the role of programmed cell deaths including apoptosis, necroptosis, autophagy and lysoptosis has been of central importance. Thus, the recent observation that immune cells from COVID-19 patients are more prone to die by apoptosis associated with disease severity [16] demonstrate the importance of these mechanisms in the context of human infectious diseases. Of interest, it was shown in vivo that inhibition of caspase prevents CD4 T cell apoptosis and Aids [17] (**Estaquier** team).

## 2. Team Work

### 2.1. Team Adriouch

Dr. Sahil Adriouch is working in the INSERM U1234 Unit located in Rouen. His current work investigates how cell death can be manipulated in the context of cancer to fuel anti-tumor immune responses. It is currently established that newly described forms of cell death, akin to pyropotosis or necroptosis, that represent catastrophic forms of cell death as opposed to apoptosis, can stimulate innate as well as adaptive immune responses. These forms of cell death have been termed “immunogenic cell death” (ICD) [18,19,20]. The idea has then emerged to manipulate the induction of ICD in the tumor context and not only kill a fraction of the tumor cells, but also and at the same time, stimulate the release of danger signals and of tumor antigens, with the aim to stimulate antitumor immune responses [20,21].

Induction of ICD is emerging as a potent trigger of anti-tumor immune responses and can be induced not only by the activation of pyroptosis executioners like gasdermin D (GSDMD) or gasdermin E (GSDME), but also by the activation of Mixed Lineage Kinase Domain-like Protein (MLKL), which can similarly oligomerize and also directly disrupts the membrane integrity to cause necroptosis [22,23,24,25]. Therefore, in the envisioned strategy, Adriouch team aims to use viral vectors, akin to AAV, to induce ICD intratumorally by expressing active truncated/mutated forms of GSDMD, GSDME or MLKL, thus bypassing tumor resistance to the induction of pyroptosis or necroptosis. Induction of tumor cell death in the context of acute inflammation and release of DAMPs in conjunction with tumor antigens is expected to enhance presentation of tumor antigens via DC to T cells and finally induce a robust, long lasting adaptive anti-tumor immunity.

Importantly, this approach can be combined easily with established immunotherapeutic approaches (e.g., immune checkpoints blockade) and the favorable safety profile of AAV vectors may facilitate the translation of such a strategy. Finally, this approach may increase tumor immunogenicity even in immunologically “cold” tumors that still represent a clinical challenge.

### 2.2. Team Brenner

The team of Dr. Catherine Brenner in UMR 9018 CNRS at Gustave Roussy, Villejuif, is exploiting the connection between apoptosis, necrosis and autophagy for cardioprotection. Dr. Brenner and colleagues showed the possibility to activate pro-survival pathways to protect from the toxicity of anticancer drugs (e.g., camptothecin, doxorubicin) without affecting their efficacy using novel chemical entities as well as repositionable molecules.

Prompted by a continuous increase in the incidence of cancer worldwide, the treatment of cancer has made tremendous progress during the last decades. Indeed, the definition of the hallmarks of cancer [26,27] has provided many druggable targets, notably within the deregulation of cell death pathways. Thus, various therapeutic approaches have been explored based for example on the inhibition of cell surface receptors by recombinant antibodies, cytotoxic compounds and chemically modified natural substances (e.g., taxanes). One promising class of anti-cancer molecules refers to the so-called small molecules developed by chemists and optimized by pharmacists for their bioactivity, solubility, and specificity. However, it appears that there are generally two major drawbacks limiting their administration: severe side effects and resistance acquisition favoring cancer relapse and metastatic spread. If small molecule toxicity can affect many organs in the body, the heart is probably one of the most problematic, because cardiotoxicity in response to various drugs and/or radiotherapy can be observed in survivor cancer patients at long term [28]. This has been particularly documented in the past for pediatric cancers treated by anthracyclines with an increased incidence of heart failure, pericardial diseases and valvular diseases [29].

Otherwise, deep cellular and molecular investigation pointed to connections between survival and cell death pathways, revealing new therapeutic perspectives. Thus, using small molecules identified by phenotypic high throughput screenings, team Brenner showed the possibility to activate pro-survival pathways to protect from the toxicity of pro-apoptotic anticancer drugs (e.g., camptothecin, doxorubicin) and H202 (as a necrosis inducer) without affecting their efficacy [30]. Thus, cell treatment with novel chemical entities as well as repositionable molecules such as the well-known cardiac glycosides (i.e., digoxin and digitoxigenin) regulated apoptotic BCL2 family members such as BCL-XL and BAX, activated selectively and Atg5 and Beclin1-dependent autophagy, stimulated mitochondrial fission, reactive oxygen species (ROS) production and metabolic reprogramming and finally inhibited cell death. Thus, by real measurement of energetic metabolism with Seahorse technology, the team observed that increasing ATP production favors the survival of cells and contributes to inhibiting cell death induction.

In conclusion, these recent findings highlight the potential translational application of interdisciplinary basic research between biologists and chemists [30]. This raises the hope to open the door to the preclinical and clinical development of novel cardioprotective molecules to prevent the long-term adverse effects of chemotherapy or radiotherapy for survivor patients.

### 2.3. Team Desagher

The team led by Dr. Solange Desagher at the Institute of Molecular Genetics of Montpellier (IGMM) has been studying the regulation of apoptosis by the E3 ubiquitin-ligase TRIM17 for several years [31].

Indeed, accumulating evidence indicates that the ubiquitin-proteasome system (UPS) regulates apoptosis by controlling the function and level of key apoptotic factors [32]. E3 ubiquitin-ligases confer a high level of specificity to this system by recognizing the target proteins and mediating their ubiquitination. They participate in apoptosis regulation by ensuring the rapid degradation of proteins that can either trigger or inhibit apoptosis. Neuronal apoptosis plays a crucial role in neurodegenerative diseases [33,34]. Therefore, key deubiquitinases and E3 ubiquitin-ligases are promising therapeutic targets for the prevention of neurodegeneration [35]. In particular, TRIM proteins, which represent one of the largest classes of RING-containing E3 ubiquitin-ligases, are potentially druggable [36].

Desagher team identified *TRIM17* as one of the most upregulated genes during early apoptosis of primary cerebellar granule neurons [37]. TRIM17 then appeared to be both necessary and sufficient for apoptosis in various neuronal types, its pro-apoptotic activity depending on its RING domain [38]. TRIM17 exerts its pro-apoptotic function in part by mediating the targeted degradation of MCL-1, an anti-apoptotic protein of the BCL-2 family protein that plays a critical role in the survival of most cell types and contributes to chemoresistance in many cancers [39]. Indeed, the team showed that TRIM17 participates in the ubiquitination and degradation of MCL-1, which is necessary for the initiation of neuronal apoptosis [40]. TRIM17 also regulates the stability of BCL2A1, another anti-apoptotic protein of the BCL-2 family but in the opposite way. Indeed, Desagher team showed that TRIM17 inhibits TRIM28-mediated ubiquitination and degradation of BCL2A1, thereby promoting the survival of BCL2A1-dependent cells, including chemotherapy-resistant melanoma cells [41]. Additional studies showed that TRIM17 can stabilize proteins other than BCL2A1 by inhibiting different E3 ubiquitin-ligases of the TRIM family. Indeed, the team found that TRIM17 inhibits the ubiquitination/degradation of transcription factors ZSCAN21 and NFATc3 mediated by TRIM41 and TRIM39, respectively [38,42]. Because these transcription factors control the expression of proteins involved in neuronal death, their stabilization may also contribute to the pro-apoptotic effect of TRIM17 [43,44]. In particular, ZSCAN21 promotes the expression of α-synuclein, whose accumulation leads to neurodegeneration [42]. The team is currently investigating the role of TRIM17, TRIM41 and ZSCAN21 in the transcriptional regulation of α-synuclein and neuronal death in Parkinson’s disease.

### 2.4. Team Devos

The multidisciplinary pre-clinical and clinical research team, led by Prof. David Devos at Lille Neuroscience & Cognition Center is studying notably the molecular and pharmacological mechanisms of neuronal death, particularly ferroptosis in Parkinson’s disease (PD) and amyotrophic lateral sclerosis.

PD has been defined by the regulated cell death of dopaminergic neurons predominantly in the substantia nigra pars compacta (SNc) associated with aggregation of alpha-synuclein (α-syn) within Lewy bodies and constant accumulation of iron. Dopaminergic neurons are naturally rich in iron because iron is essential for the synthesis and metabolism of dopamine and the mitochondrial aerobic metabolism.

For many years, the main process studied in neuronal death was apoptosis. This was due to the fact that only a few types of programmed cell death were known, and they were identified mainly using oncogenic cell lines such as neuroblastomas [45,46]. Since then, several types of cell death have been discovered. Ferroptosis has been established as a novel form of regulated necrosis that is clearly distinct from other known cell death pathways [47]. Ferroptosis is characterized by iron-dependent lipid peroxidation that massively decreases under iron chelation. From the early days of its characterization in 2012, the Devos team has identified several pathological features of PD as key components of ferroptosis [46,48,49,50,51,52,53]. They published the first work clearly showing that ferroptosis was prevalent in both in vitro and in vivo models of PD [54]. Furthermore, the team demonstrated that the action of sporadic PD-associated neurotoxins could be counteracted by specific ferroptosis inhibitors such as ferrostatin-1, liproxstatin-1 and iron chelators [54]. They also showed with a pilot clinical trial that the iron chelator deferiprone may be able to slow down PD and ALS progression [48,55]. This led to obtaining European H2020 funding to perform a therapeutic trial of deferiprone on 372 patients in 23 centers in 8 European countries (https://www.fairpark2.eu/ accessed on 21 June 2022).

The current work in Devos team shows that the amount of α-syn modulates sensitivity to ferroptosis and polyunsaturated fatty acid lipid profiles and the amount of iron sensitizes to ferroptosis even without the classical PD neurotoxin. All these molecular demonstrations lead to several therapeutic developments in collaboration with pharmaceutical companies and drug discovery patents.

### 2.5. Team Dubrez

Knowledge of molecular mechanisms that orchestrate apoptosis grew considerably in the late 90s following work in *Caenorhabditis elegans*. Most of the molecular actors that directly participate in or regulate programmed cell death have been discovered within a 10 year period. In the early 2000s, their functional characterization expanded the scope of our understanding. Many effectors or regulators of cell death have been shown to also have non-apoptotic functions. Along those lines, the research project led by Dr. Dubrez arose from the observation that apoptotic caspases and the cellular inhibitor of apoptosis 1 (cIAP1) can regulate cell differentiation [56,57]. Contrary to X-linked IAP (XIAP) that can bind apoptotic caspases and directly block their protease activity [58,59,60], the ability of cIAP1 to inhibit caspase activity has long been controversial. The debate was definitively closed in 2006 by biochemical and structural studies demonstrating that the XIAP amino acid residues required for caspase inhibition are not conserved in cIAP1 and cIAP2 [61]. The Dubrez team investigates the role of cIAP1 in signaling pathways driving cell differentiation and proliferation. Unexpectedly, cIAP1 is mainly, if not exclusively expressed in the nucleus of hematopoietic stem cells. Such a nuclear localization was also detected in some cancer cells [57,62]. Thus, the team initiated a research project aiming to explore the nuclear functions of cIAP1. Since cIAP1 displays E3-ubiquitin ligase activity, the work of the Dubrez team focuses on identifying novel ubiquitination substrates that can account for its proliferative, differentiating and oncogenic activities.

### 2.6. Team Estaquier

The team led by Dr. Jerome Estaquier at the INSERM U1124 (Paris) and at Laval University (Quebec) has been dedicated to study the mechanisms of programmed cell death (PCD) in the context of host–pathogen interactions. Thus, the specificity of the team is upstream research aimed at clarifying the biochemical and molecular mechanisms of PCD associated with immune responses, such as the role of death receptors and of mitochondria, and analyzing these mechanisms in vivo by using as a model of human diseases non-human primates. One of the main contributions arising in HIV field was the demonstration that apoptosis discriminates pathogenic and non-pathogenic infections correlating with Aids progression. They also showed the role of Fas and its ligand (FasL) in the death of memory CD4 T cells from HIV patients. In term of therapeutic approaches, the team also demonstrated the role of cysteine proteases (caspases) during Aids and that the administration of a broad caspase inhibitor prevents CD4 T cell depletion and delays Aids in non-human primates [17]. Thus, caspase inhibitor may represent an adjunctive therapeutic agent to control HIV infection. The team also recently demonstrated the importance of T cell apoptosis associated with the severity of COVID-19, underlying the role of immune cell death as a central event in the pathogenesis of viral infections [16].

In the context of intracellular pathogens, Estaquier’s team demonstrated the role of the mitochondria as a hub in controlling host–pathogen interactions. Thus, manipulation of host metabolic fluxes by pathogens represents a strategy to circumvent host immune response leading to long-term parasite survival and plays an important role in the pathology of infection. In this context, several viruses were shown to modulate pro-apoptotic Bcl-2 members and mitochondria permeabilization. In this way, mitochondria dynamics were shown to be crucial in regulating mitochondria permeabilization and immune defense.

Thus, the team is in front of the line for understanding PCD in the context of host-pathogen interactions proposing novel strategies to improve immune responses against microbes.

### 2.7. Team Gibert

Dr. Benjamin Gibert is a CNRS researcher at the CRCL (Lyon), where his team investigates novel biotechnological agents, in particular internal radiotherapy, for inducing cancer cell death.

It has been reported that proteins of the Netrin family play a crucial role during the development of the central nervous system and in various pathologies [63,64]. In particular, the Netrin-1 protein which is the ligand of several apoptosis-inducing dependence receptors is described as a key therapeutic target in many types of cancer and an anti-netrin-1 therapeutic antibody is currently being tested in the clinic [14,65,66]. Surprisingly, and probably due to the lack of molecular tools, the biology of the other members of the netrin family, namely netrin-3, -4, -5 -G1 and -G2, has not been studied, except for some brief descriptions of netrin-4 [67]. Netrin-3 protein, first described in 1999 for its putative role as a guidance molecule, has never been studied in detail for its role both in development and pathological contexts [68]. Interestingly, team Gibert showed that Netrin-3 expression correlates with cancer aggressiveness and poor patient prognosis for both neuroblastoma and small cell lung cancer (SCLC) [69]. Of note, SCLC is the most lethal histological subtype of lung cancer and is associated with high rates of metastatic disease at time of the diagnosis. This relatively rare tumor accounts for 10% of all lung cancers, but no new treatments have been discovered since the 1970s and the standard of care remains chemotherapy. The team showed that Netrin-3 has a high specificity for SCLC cells and its therapeutic targeting using a blocking antibody reduces cancer growth. This suggests that Netrin-3 could represent a therapeutic vulnerability for SCLC. Much work remains to be done to understand the molecular mechanisms and clinical implications of axonal guidance factors in cancer and resistance to treatment via cell death blockade in particular in neuroendocrine tumors [70,71].

### 2.8. Team Gillet

Dr. Germain Gillet leads a research team at the Cancer Research Center of Lyon (CRCL). In 1995, Dr. Gillet and colleagues demonstrated that the *nr-13* gene, one of the few *bcl-2* homologs known at that time, plays a key role in the anti-apoptotic effect of the *v-src* oncogene [72]. Since then, his team made seminal contributions in the field of cell death by showing that the non-canonical roles of Bcl-2 proteins, in particular regarding Ca^2+^ trafficking, are critical for the control of cell movements during the initial steps of vertebrate development [73,74]. More specifically, their data suggest that Bcl-2 family proteins control cytoskeleton dynamics and early embryonic cell movements, independent of apoptosis, by maintaining physiological intracellular Ca^2+^ shuttling between the endoplasmic reticulum (ER) and the mitochondria.

The Gillet team recently provided convincing evidence that such functions have been conserved throughout evolution. They characterized the Bcl-2 family in *Trichoplax adhaerens*, the most primitive metazoan known to date, and demonstrated that the basics of apoptosis control are conserved from *T. adhaerens* to mammals. Moreover, they found that peptides derived from Trichoplax Bcl-2 homologs can interact and inhibit human Bcl-2 proteins, sensitizing cancer cells to chemotherapy [75].

In development, Dr. Gillet and colleagues currently focus on the contribution of the non-canonical roles of Bcl-2 family proteins to shape the embryo. Using genetically engineered mouse and zebrafish models, they currently analyze the molecular mechanisms by which Bcl-2 family proteins influence cell survival and migration. They intend to (i) identify the signaling networks that lead to the modulation of Ca^2+^ homeostasis and cell movements by Bcl-2 family proteins, (ii) identify the factors involved in apoptosis progression and cell migration and (iii) review the roles played by these factors, depending on the physiopathological context.

Regarding oncogenic transformation, the team focuses on Nrh, a Bcl-2 homolog also referred to as Bcl2l10 or Bcl-B, which is over-expressed in breast cancer (BC). Actually, there is evidence that disrupting Nrh/IP3R1 interactions at the level of the ER may be a promising strategy to prime BC cells to death [76]. The team works on confirming the prognostic and/or predictive value of Nrh expression levels while validating the therapeutic potential of compounds targeting the Nrh/IP3R1 complex. This work may lead to the identification of novel prognosis markers and deliver potential molecular targets for inhibiting tumor growth and metastasis onset.

It is anticipated that this research will continue to open new avenues about the actual functions of the Bcl-2 family proteins, in the context of development and tumorigenesis.

### 2.9. Team Guenal-Mignotte

The Stress and Cell Death team initially led by Pr. Bernard Mignotte and now by Pr. Isabelle Guénal was one of the first to point out the role of mitochondria in apoptosis [77].

In mammals, it is now well established that Bcl-2 family proteins control mitochondrial outer membrane permeabilization leading to the release of pro-apoptotic factors in the cytosol. This critical step of the so-called mitochondrial apoptosis pathway appears dispensable in some organisms such as *C. elegans* and *D. melanogaster*. However, evidence accumulates in favor of a mitochondrial involvement in apoptosis in these organisms. In *C. elegans*, mitochondria have been shown to be a sequestration site for Ced-9/Ced-4 complexes, thus preventing caspase activation as long as the BH3-only protein Egl-1 is not present, and mitochondrial proteins such as endonuclease G are involved in the execution of cell death [78]. In *Drosophila*, among pro-apoptotic factors, RHG proteins that promote Diap1 degradation [79] and the two identified Bcl-2/Ced-9 family members constitutively or transiently localize at the mitochondria [80].

The team now studies the role of mitochondria in cell death processes in *Drosophila*. One may ask why they are studying Bcl-2 family protein-dependent apoptosis in *Drosophila*, a model organism in which permeabilization of the outer membrane of mitochondria is unnecessary? In fact, this particularity seems to be an advantage for at least three reasons: (i) it is of fundamental interest from an evolutionary point of view; (ii) it allows the identification of Bcl-2 family protein activities that are potentially masked by the permeabilization of the outer mitochondrial membrane and (iii) the well-established power of the genetic approaches in *Drosophila* facilitates the identification of genes regulating these activities *in vivo*. The team showed that *Drosophila* Bcl-2 family members participate in the cell death process at different levels, such as mitochondrial ROS accumulation and mitochondrial network fragmentation [81,82]. Using Rbf1, the fly retinoblastoma protein homolog as an inducer of Bcl-2-family-protein-controlled-apoptosis, the team identified a specific JNK pathway inducing apoptosis-induced proliferation [7] and showed that the interaction between Debcl (the *Drosophila* pro-apoptotic Bcl-2 family member) and the fission protein Drp1 is required for apoptosis. They also showed that the endoplasmic reticulum (ER) protein Buffy (the only *Drosophila* anti-apoptotic Bcl-2 family member) is the main regulator of this interaction. *Drosophila* is therefore a powerful model to study the involvement of Bcl-2 family members and other mitochondrial proteins in cell death processes in vivo.

Current work in the team focuses on mitochondrial dynamics, mitochondrial quality control and mitochondria-ER dialog during apoptosis and ferroptosis.

### 2.10. Team Ichim

The team led by Dr. Gabriel Ichim at the CRCL in Lyon investigates how cell death fuels several oncogenic processes such as proliferation, migration and invasion.

It is currently well established that apoptosis is a roadblock for oncogenic transformation by efficiently removing transformed cells. Accordingly, cancer and apoptosis seem to be in a perpetual antithetic game: apoptosis eliminates cancer cells, while cancer constantly develops strategies to evade apoptosis [26]. However, the Ichim team is driven by the conviction that oncogenesis hijacks apoptosis and its effectors, caspases and mitochondria to fuel certain hallmarks. Indeed, the team showed that low-level caspase activation is compatible with cancer cell survival [11,83]. Notably, this state of failed apoptosis promotes melanoma aggressiveness [83]. Despite this, the role of caspases aside from their canonical function in apoptosis is rarely investigated in oncogenesis and team Ichim is currently developing two projects aiming to narrow this knowledge gap.

First, there is extensive research on how caspase lethal activation is finely regulated by cell-autonomous buffering mechanisms such as IAPs that are overly activated in cancer to circumvent cell death [84,85]. However, very little is known on how the tumor microenvironment impacts on caspase activation. As the tumoral stiffness and the inherent mechanical stress gain notoriety in favoring oncogenesis, one could wonder if this pro-oncogenic effect is partly explained by an unknown inhibitory effect of mechanical stress on caspase activation and efficient induction of tumor cell death [86,87]. The team is currently investigating this and it has preliminary data showing that mechanical stress, similar to that encountered during metastasis, restricts lethal caspase activation, which is permissive for cancer cell survival and metastasis [88].

Secondly, therapeutic strategies based on lethal caspase activation are sometimes ineffective, since caspases are not entirely anti-tumoral and might have apoptosis-unrelated roles [2,11]. This is in line with several studies describing high levels of caspase-3 in melanoma, although no plausible explanation was given for its possible functions [89,90]. Team Ichim therefore explores whether caspase-3 has a non-canonical role on cancer aggressiveness, irrespective of apoptosis execution, with a special emphasis on cancer cell motility.

### 2.11. Team Juin

Dr. Philippe P Juin leads a research team called Adaptation and Tumor Escape in Breast Cancer at the Nantes-Angers Cancer and Immunology Research Center. Their research objectives are to (i) describe the molecular mechanisms and reprogramming pathways that, in response to various types of cellular stress, participate in the organization of epithelial tumors as a pseudo-organ; (ii) to understand the signals involved in the escape from tumor suppression and (iii) to identify therapeutic approaches to reactivate this suppression. The team focuses their research on breast carcinoma, albeit non-exclusively.

To reach its objectives, right now the Juin team is exploring different leads. First, they characterize the biochemical and cellular determinants of mitochondrial priming/processing in individual cells and in populations [91,92,93,94,95,96]. Secondly, the Juin team assesses the role of mitochondrial stress and cell death on the organization of epithelial diversity and the influence of primary ciliogenesis in cancer stem cell resistance [91,92,93,94,95,96]. Thirdly, they study the reciprocal influences between heterotypic intercellular communications in the microenvironment, mitochondrial signaling and cell survival. Finally, the team maps the epigenetic events related to mitochondrial signaling and cell survival in tumor ecosystems.

### 2.12. Team Kroemer

The Kroemer team is working on several aspects of human pathophysiology, placing emphasis on how intracellular stress is communicated to the extracellular world or vice versa, how environmental or organismal challenges elicit cell-autonomous stress responses.

In the area of cell death research, this team is particularly interested in immunogenic cell death (ICD), which turned out to play a fundamental role in the success of cancer treatment by chemotherapy, radiotherapy or targeted therapies. Indeed, it had been thought that apoptosis would be an immunologically silent cell death modality, while necrosis would be pro-inflammatory and hence immunogenic. However, this dogma turned out to be wrong (or only partially correct) because anthracycline-killed cancer cells lose their capacity to elicit an antitumor immune response if pro-apoptotic caspase are inhibited causing a switch from apoptosis to necrosis [18]. Since this initial discovery, the Kroemer team has been defining the danger-associated molecular patterns (DAMPs) that are released or exposed by dying cancer cells to act on pattern recognition receptors (PRRs) and stimulate an innate (and later cognate) immune response. Schematically, there are two types of DAMPs that explain ICD.

First, several DAMPs are reduced by cancer cells as soon as they die, irrespective of the precise mechanisms of cell death. This applies to the cytosolic protein annexin A1, which acts on formyl peptide receptor-1 (FPR1) on dendritic cells (DCs) [97], as well as to the nuclear protein high mobility team B1, which acts on Toll-like receptor-4 (TLR4) on DCs [98].

Second, other DAMPs are only produced by cancer cells if they manifest a specific set of premortem stress responses such as autophagy (which facilitates the release of the DC-chemoattractant ATP) [99,100], the integrated stress response secondary to the inhibition of DNA-to-RNA transcription (which stimulates the plasma membrane exposure of the ‘eat-me’ signal calreticulin) [101,102] and a type-1 interferon response (which facilitates the release of chemokines attracting T cells into the tumor microenvironment) [103].

Of note, failure to emit these DAMPs or to perceive them via PRRs results in reduced immune surveillance as well as in failing responses to cancer chemotherapies, in mouse models as well as in cancer patients [104,105]. The Kroemer team is now identifying novel pharmacological ICD inducers and exploring new strategies to enhance the response of anticancer immune effectors.

### 2.13. Team Legembre-Vacher

Dr. Patrick Legembre (UMR CNRS 7276, Limoges) and Dr. Pierre Vacher (INSERM U1045, Bordeaux) worked together for many years to understand how calcium homeostasis affects the signaling pathways triggered by CD95 (also known as Fas). CD95 is a member of the tumor necrosis factor (TNF) receptor family and is considered as a prototype of the death receptors. Its cognate ligand, CD95L (FasL) is a transmembrane ligand that belongs to the TNF superfamily. CD95L can be cleaved by metalloproteases to release a soluble and non-toxic soluble CD95L (s-CD95L). Although the membrane-bound CD95L (m-CD95L), which is mainly expressed by activated T cells and natural killer cells, induces an apoptotic death in infected and transformed cells, s-CD95L fails to do it and instead implements non-apoptotic signaling pathways (i.e., PI3K, NFkB). By doing so, s-CD95L contributes to the metastatic dissemination of cancer cells [106] or the inflammatory process in lupus patients [107]. Furthermore, recent data revealed that independently of its ligand, the expression of CD95 itself in triple negative breast cancer (TNBC) cells controls the NFkB signal [108] and thereby, might prevent the NK-mediated anti-tumor response [109].

Because CD95 engagement induces complex Ca^2+^ signaling pathways which inhibit apoptosis [110] and promote the non-apoptotic signals [107,111], the team wondered whether and how ion channels can affect the above mentioned apoptotic and non-apoptotic signals in immune and TNBC cells. Several ion channels (i.e., chloride, potassium…), most of them being calcium-dependent, were shown to be involved in NFkB/inflammasome signaling [111,112,113]. Another question that the team would like to address is whether the CD95-mediated inhibition of NF-kB is a mechanism only observed in transformed cells or whether this molecular mechanism can also occur in normal cells.

In summary, although the “classical” tumor evasion associated with the loss of CD95 expression by tumor cells exposed to CD95L-expressing immune cells remains valid, this oncogenic process might be counterbalanced by the induction of an NF-κB-driven pro-inflammatory response observed in tumor cells losing CD95. Indeed, this latter signal ultimately stimulates the NK-mediated anti-tumor response [109].

### 2.14. Team Levayer

Dr. Romain Levayer leads a research team at the Institut Pasteur where he focuses on the fine tuning of cell death in epithelia using a combination of live imaging, modeling, genetics and optogenetics in *Drosophila*.

Cell elimination by apoptosis is essential for tissue morphogenesis and adult tissue homeostasis. While the core pathway of apoptosis has been well characterized, how the elimination of cells is coordinated locally and at the tissue level remains largely unknown. For instance, very little is known about the processes regulating the distribution of cell death in time and space, as well as the total number of cells that will die in a given tissue. Similarly, how local perturbations such as wounds, overgrowth or aberrant cell fate can impact the survival/death of neighboring cells remains poorly understood. This plasticity is well illustrated by the concept of cell competition: the context-dependent elimination of suboptimal cells triggered by the vicinity with wild type (WT) cells [114]. Recently, Levayer team and others have shown that the mechanical stress generated by growth could trigger preferential elimination of one cell population if it is more sensitive to mechanical pressure [115,116]. This process called mechanical cell competition could fasten the expansion of pre-tumoral cells by eliminating neighboring WT cells through apoptosis [117]. Levayer team is currently characterizing the mechanosensitive pathways sensing deformations and regulating caspase activation [118] while defining the minimal conditions required for mechanical competition.

Interestingly, the pathways involved in mechanical competition may also play an essential role for the spatio-temporal coordination of cell eliminations in physiological contexts. For instance, the team recently found that apoptotic cells in *Drosophila* epithelia trigger a transient protection of their direct cell neighbors against apoptosis through a short activation of EGFR/ERK [119], a pathway that we previously identified for its function during mechanical cell competition [118]. This transient protection is essential to disperse cell death in time and space and avoid elimination of cells in clusters, a condition highly detrimental for tissue sealing properties. A similar transient protection was characterized in MCF10A cells and in HeLa cells [120]. These works and others [121] highlight the essential collective effects at play for the regulation of apoptosis in epithelia and the self-organized properties emerging from the integration of multiple spatial feedbacks. Levayer team is currently dissecting the contribution of these various feedbacks (positive and negative) to build a more predictive framework of apoptosis regulation at the tissue level.

### 2.15. Team Manon

The Bcl-2 family proteins regulate the permeability of the outer mitochondrial membrane to apoptogenic factors that are released from mitochondria at the early steps of apoptosis. They exhibit some functional redundancy, such as between Bax and Bak, Bcl-2 and Bcl-xL or Bim and Puma. They follow at least two distinct, non-exclusive, modes of regulation that are termed direct (for example, the direct activation of Bax by Bim) or indirect (for example, the release of the interaction between Bax and Bcl-2 by Bad).

These intertwined interactions complicate the studies in mammalian cells, where it is not always possible to assess the function of a unique Bcl-2 family member independently from the rest of the network. Investigators have developed simplified systems, such as in vitro reconstituted models, where recombinant proteins are tested on isolated mitochondria or artificial liposomes. These models have provided an outstanding amount of knowledge about the molecular aspects underlying the function of Bcl-2 proteins. However, they are limited by the fact that these studies are done outside of a cellular context.

An alternative approach has been to express these proteins in cells where they are not present, but where their targets, i.e., mitochondria, are. The yeast *Saccharomyces cerevisiae* was initially used as a simple tool to test the interactions between Bcl-2 family members, through the widely popular and relatively easy-to-use two-hybrid system [122]. Serendipitously, investigators observed that Bcl-2 family members largely conserved their ability to interact with mitochondria and to modulate the permeability of outer mitochondrial membrane when expressed in yeast.

This led the Manon team to develop studies in yeast expressing the pro-apoptotic protein Bax, to gain information on how Bax interacts and permeabilizes the outer mitochondrial membrane [123]. Furthermore, they contributed to the investigation of new Bax regulations by yeast homologs of mammalian proteins, such as the mitochondrial protein receptor Tom22 [124] and the protein kinase AKT/Sch9 [125].

The team is now interested in defining the role of mitochondria-ER contact sites in the process of translocation versus retro-translocation of Bax and its partners [126]. Owing to the presence of the yeast ERMES complex that regulates the stability of these contacts, they will get a better knowledge of how they modulate the mitochondrial localization of Bax (and other Bcl-2 family members) and, consequently, apoptosis.

### 2.16. Team Mehlen

Dr. Patrick Mehlen is the director of the CRCL in Lyon and a CNRS researcher. His team investigates mechanisms and therapeutic implications of cell death with a large focus on cancer.

His team has over the years developed the notion of Dependence Receptors (DRs) [127]. These particular receptors form a functional family of transmembrane receptors able to display two opposite signals depending on ligand availability. While upon binding of their respective ligands, these receptors trigger various signals, upon the absence of ligands they actively trigger cell death [128]. Their expression thus induces a state of cell dependence for survival on the presence of the ligand in the cellular micro-environment. While they have pleiotropic roles during embryonic development, they often have been described as tumor suppressor genes as their presence at the plasma membrane is a constraint for cancer progression. The team has thus demonstrated that the ligands of these DRs are often overexpressed in different tumor types as a selective mechanism to block DRs-induced apoptosis [129]. This has been extensively described for the receptors DCC and UNC5B that bind netrin-1 and it has been shown that netrin-1 interference is associated with tumor growth inhibition in various models [130]. A therapeutic strategy based on the disruption of the receptor/ligand interaction has thus been proposed and anti-netrin-1 monoclonal antibody has been preclinically [131] and clinically developed. This antibody has shown an excellent safety profile and signs of clinical efficacy in a phase 1 trial and a series of phase 2 trials are ongoing testing the anti-netrin-1 mAb in combination with immune checkpoint inhibitors or chemotherapies in different clinical indications. In parallel to the clinical activity, the Mehlen team is trying to define further the precise mode of action of the netrin-1 antibody considering its impact on cancer cell death and phenotypic plasticity. In addition, basic and translational research is done on other ligand/DRs pairs to explore the role of these DRs in the control of tumor progression.

### 2.17. Team Meurette

Dr. Olivier Meurette is a lecturer at the pharmacy faculty of Lyon and leads a research team at the CRCL in Lyon, where he investigates the molecular link between autophagy and regulated cell death, as a target for cancer therapy.

Regulated cell death is now recognized to adopt different phenotypes and follow different signaling pathways, with many interconnections [132]. The crosstalk between these different signaling hubs may be an actionable target for cancer therapy and circumvent the barriers established by cancer cells to escape physiological cell death. In particular, autophagy–apoptosis crosstalk is a major determinant of life and death decisions [133] and of chemotherapy efficiency [134]. The team is studying how dependence receptors may induce autophagy-dependent cell death and the molecular pathway involved in this pathway. More specifically, Kremen-1 dependence receptor [135] is inducing both apoptotic and autophagic features upon its overexpression in cancer cells. This cell death is dependent on ATG5 and dramatically reduces when the autophagic flux is inhibited prior to autophagosomes formation. However, cell death is fostered when autophagic flux is inhibited downstream of autophagosomes formation. The team therefore attempts to decipher how autophagosomes accumulation and crosstalk with apoptosis may lead to cell death. The team also adopts genome-wide screening strategies to characterize signaling pathways involved in Kremen1-induced cell death, as well as identify small molecules specifically triggering aberrant and lethal autophagy in cancer cells.

### 2.18. Team Micheau

The team headed by Dr. Olivier Micheau studies the mechanisms of early membrane pro-apoptotic signal transduction, also coined extrinsic pathway. His team has mostly focused on understanding how pro-apoptotic agonist receptors of the TNF superfamily initiate cell death [136,137,138,139]. These include Fas, TNFR1, DR4 or DR5, whose engagement by cognate ligands can lead to activation of cell death. These membrane receptors share a common protein-binding motif, the so-called death domain, which is essential and sufficient for the recruitment of adaptor proteins, initiator caspases or kinases such as the caspase-8 or RIPK1, allowing initiation of apoptosis or necrosis, respectively. For this reason, it was long thought that the initiation and regulation of cell death would occur exclusively at the plasma membrane [137,140]. It turns out that regulation and initiation of these pro-apoptotic, necrotic or even non-apoptotic signal transduction machineries, are much more complex than expected. For instance, whereas the canonical ligand-mediated Fas, DR4 and DR5 engagement initiates apoptosis directly from the plasma membrane, the pro-apoptotic machinery of TNFR1 is initiated in the cytosol, by a secondary complex [139]. Complex II arises from complex I, a membrane-bound complex dedicated to the activation of NF-kB, which, amongst other things induces the transcription of c-FLIP [141], the most potent cellular caspase-8 inhibitor [142]. As a result, and unless the NF-kB pathway is defective, TNFR1 is most of the time unable to induce cell death, contrary to Fas, DR4 or DR5. Mirroring TNFR1 complex I and II, it has been proposed that TRAIL non-apoptotic signal transduction would occur through a cytosolic complex [143,144]. If progress has been made to understand how these pleiotropic signaling capabilities are orchestrated from these receptors, the picture is still not fully understood, precluding in-depth comprehension of their physiological relevance and exploitation in the clinic.

In particular, albeit TRAIL has been and still continues to be of major interest in oncology, due to its selective anti-tumor properties and its role in anti-tumor immunity [145,146,147], clinical trials aiming at targeting DR4 or DR5 have been so far unsatisfactory [148,149]. Micheau’s team and other teams have gathered important pieces of work demonstrating that the use of TRAIL or TRAIL derivatives in the clinic still requires investigating the most fundamental aspects of TRAIL signal transduction machinery. Thus, efforts are being made in Micheau’s team to understand more precisely how and under which circumstances DR4 and DR5 agonist receptors (1) induce a pro-motile/metastatic signaling pathway in tumor cells, (2) trigger apoptosis from within the cell, in a ligand-independent manner [136] and (3) require glycosylation to efficiently signal apoptosis upon engagement of TRAIL- or FasL/CD95-induced [136].

Micheau’s team is convinced that resolving these issues will likely lead to better therapeutic strategies aiming at targeting TRAIL receptors or other TNF family receptors or ligands.

### 2.19. Team Mollereau

Pr. Bertrand Mollereau is the head of the Regulated Cell Death and Genetics of Neurodegeneration team at the Laboratory of Biology and Modelling of the Cell (LBMC) at the Ecole Normale Supérieure de Lyon (ENSL). While apoptosis is clearly the most well studied form of cell death, in the last two decades additional forms of cell death, including cell necrosis, have been molecularly characterized [150]. The understanding that behind the term of “cell necrosis” coexist several genetically controlled forms of cell death, raised important questions on how these different forms of regulated cell death integrate for the elimination of cells in different cellular and organismal contexts [12,133,151]. For example, it is intriguing that apoptosis is the main cell death mechanism for the elimination of excess cells during development, while other regulated cell death pathways are used during neurodegeneration or the death of cancer cells that are resistant to apoptosis. Team Mollereau uses *Drosophila* models of neurodegeneration and of developmental cell death to investigate the mutual interplay between apoptosis and regulated necrosis or cellular survival pathways, such as autophagy and proliferation [152,153,154,155]. His team and others characterized the antagonistic relationship between apoptosis and autophagy [133,156,157]. Specifically, they reported that activation of a protective autophagy pathway induced by a non-lethal ER stress activation inhibits apoptosis in *Drosophila* and mouse models of neurodegeneration [158,159]. Presumably, the inhibition of apoptosis by autophagy could occur by sequestration of mitochondria by mitophagy or caspases in autophagosomes, hence inhibiting cytochrome c release and caspase-mediated cleavage of cellular substrates [160,161]. They also reported a reverse inhibition of caspases on autophagy, in which the caspases Dronc, Dcp-1 and Drice inhibit autophagy flux during the apoptotic elimination of photoreceptor neurons [162]. The team also investigated the interplay between caspases and regulated necrosis, the latter being used for germ cell elimination during spermatogenesis in *Drosophila* and mice [163]. They found that germ cell necrosis requires p53 (p53B) isoform in *Drosophila* and Tp53 in mice. In addition, the dampening of basal caspase activity enhanced germ cell necrosis [164], suggesting that caspases inhibit regulated germ cell necrosis by cleaving positive regulators of necrosis.

In conclusion, the negative feedback of caspases on regulated necrosis or autophagy suggests that non-apoptotic caspase activation is responsible for the cleavage and therefore inhibition of positive regulators of these pathways. To elucidate the mechanisms by which apoptosis inhibits other concurrent cellular survival (autophagy) or death pathways (regulated necrosis), the Mollereau team is currently seeking to characterize caspases substrates that act as molecular switches between apoptosis and regulated necrosis.

### 2.20. Team Priault

The non-apoptotic functions of Bcl-2 and Bcl-xL have been the focus of interest of Dr. Muriel Priault since she was a postdoctoral researcher in François Vallette’s laboratory in Nantes (France) between 2003 and 2006. She discovered then that both Bcl-2 and Bcl-xL stimulate cell survival by other means than antagonizing apoptosis, namely by allowing cells to exhibit a stronger autophagic response when confronted by nutrient starvation [165]. This finding did not concur with the widely spread view that Bcl-2 and Bcl-xL bind and inhibit BH3-containing the autophagic protein Beclin. She pursued this topic of investigation when she became a permanent CNRS researcher in Stéphen Manon’s team in Bordeaux. She then discovered that when Bcl-xL undergoes a spontaneous modification called deamidation, autophagy is further stimulated in mammalian cells [166].

For the past 15 years, she has dissected the intrinsic instability that drives Bcl-xL deamidation. She pioneered the description of a monodeamidated form of Bcl-xL, and of the sequential mechanism that leads to the doubly deamidated form [167,168]. She has now specialized her lab on protein instability and molecular aging [169], and uses Bcl-xL deamidation both as a tool and as a study object.

The basic research axis of the Priault team aims to determine how deamidation affects Bcl-xL survival functions in cancer cells, and its tumorigenic activity after xenografts in mice. In addition, the Priault team also transfers the knowledge earned on Bcl-xL deamidation to applied science, and a patent has been filed for the use of Bcl-xL deamidation status to discriminate between central and peripheral causes on thrombocytopenia.

### 2.21. Team Ricci

Dr. Jean-Ehrland Ricci set up the Metabolism, Cancer and Immune Response (mCARE) team at INSERM U1065, C3M in Nice. His team investigates how cellular metabolism may modulate cell death and therefore cancer immuno-surveillance.

The general approach in cancer research is to define and then exploit the molecular requirements that distinguish tumor from normal cells. One hallmark of cancer cells is an exacerbated metabolism supporting biomass accumulation and controlling its redox status (so called Warburg effect). The working hypothesis of the Ricci team is that cancer cells not only have a specific energy metabolism, but they have also developed ways to prevent cell death, escape from immuno-surveillance and resist chemotherapy in order to proliferate.

Dysfunction of cell death programs can alter the homeostasis of specific cell populations and contribute to various pathological conditions, ranging from cancer to neurodegenerative diseases. During the last decades, efforts were mainly concentrated on understanding the molecular mechanisms of apoptosis, a form of cell death characterized by the activation of caspases. However, within a short time of identifying caspases as the enzymes that orchestrate apoptotic cell death, it became apparent that inhibition of caspase activity is frequently not able to preserve cell survival, even if the features of apoptosis are effectively blocked [170].

Importantly, this non–apoptotic cell death could be blocked by Bcl-2 [171]. In general, dying cells under the conditions just described do not resemble apoptotic cells and accordingly this form of cell death has been called “caspase-independent cell death” (CICD) [172]. However, a precise molecular definition of this phenomenon still remains to be found.

Ricci team is driven by the conviction that the way a cell is dying in vivo will impact on the ability of the immune system to get activated or not, eventually leading to enhanced immuno-surveillance. They have demonstrated that modulating the cancer cell metabolism, using glycolytic inhibitors or diet, can impact on the response to targeted chemotherapies and enhance the anti-cancer immune response [173,174,175]. The Ricci team clarified the regulation and the importance of CICD in cancers [176,177] and uncovered how metabolism may impact on the function of B and T lymphomas [178,179,180]. Finally, they are deciphering how the main powerhouse of the cells, the mitochondria, is recycled in cancer cells and how it may modulate the response to chemotherapies [181,182].

The Ricci team therefore focuses on the molecular basis of tumor metabolism in cancers—how it can be targeted and how it may influence cell death and the anti-cancer immune response—in a continuum from fundamental mechanisms to clinic.

### 2.22. Team Poyet

Dr. Jean-Luc Poyet leads the “Protein–protein interaction in the control of apoptosis” team in the IRSL in Paris. The team’s work aims to design and develop peptidic modulators of protein–protein interactions that act on key apoptosis regulators.

Although the past two decades have witnessed tremendous progress in anticancer drug development, cancer still poses a major threat to public health worldwide. Several pathways, implicating a large variety of tumor survival and tumor suppressor genes, have been identified in the development of tumors and most current therapeutics essentially target these oncogenic signaling networks. However, intrinsic or acquired resistance often limits the efficacy of these targeted therapies. Due to their basal stress phenotype associated with oncogenic transformation, cancer cells are also addicted to non-mutated, non-oncogenic proteins that do not perform such vital functions in normal cells. Targeting these non-oncogene addictions in the context of a cancerous phenotype could therefore induce selective killing of cancer cells, opening a number of interesting therapeutic opportunities.

Anti-apoptosis clone-11 protein (AAC-11) is an apoptosis-inhibiting nuclear protein highly expressed in various cancer cells and tissues, this overexpression being associated with poor prognosis [183,184,185]. Recent studies revealed that AAC-11 expression correlates with anticancer drug resistance and contributes to tumor invasion and metastasis [186,187,188,189]. Moreover, AAC-11 acts as an immune escape factor, conferring tumor immune resistance to antigen-specific T cells [190]. These observations make AAC-11 a significant player in cancer cell progression, survival and spread. Therefore, its inactivation might constitute an attractive approach for developing cancer therapeutics. Team Poyet has developed cell-penetrating peptides derived from AAC-11 that selectively disrupt vital cellular functions and induce apoptosis in a plurality of cancer cells through the inhibition of protein–protein interactions between AAC-11 and its partners, while sparing normal cells. These mimetic peptides are based on the fusion of a cell penetrating sequence and portions of the leuzine-zipper domain of AAC-11, which functions as a protein–protein interaction module. These so-called LZDPs (Leuzine-Zipper Derived Peptides) have demonstrated promising therapeutic benefits in various mouse models of cancer (i.e., melanoma, triple-negative breast cancer, Sézary syndrome and acute leukemia) with limited toxicity [191,192,193,194,195]. Interestingly, cancer cells death mediated by the peptides is immunogenic and provides efficient anti-tumor immunity in different tumor vaccination models. Team Poyet now explores the development of the LZDPs as novel anticancer agents.

### 2.23. Team Riquet

Following his appointment in October 2013 at Ghent University as visiting research professor, Dr. Franck B. Riquet, associate professor at Lille University since 2009, has established the Death Dynamics Team (DDT) within the “Molecular Signaling and Cell Death” research unit led by Prof. P. Vandenabeele at the Center for Inflammation Research in Belgium. The DDT currently operating from both university campuses investigates the regulation of cell fate decision (balance between cell survival and cell death) and its immunogenic consequences, focusing on determining how cell signaling events are encoded and computed at the biochemical level, in real-time and at the single living cell level. To this end, the DDT is now also developing its research activities within the ‘PhysBio’ team part of the “Dynamique des Systèmes Complexes” (Dysco) subunit at UMR 8523-PhLAM.

At the cellular level, biological cues are processed and encoded through the spatial and temporal biochemical signaling dynamics of signaling networks. The information is then decoded to elicit specific cellular processes. In recent years, cell biology has revealed a new level of complexity in signaling, in which the cell encrypts information based on temporal modulation of its signaling activities [196,197]. ERK1/2 signaling has been an early and classic example of how signaling dynamic features drive cellular responses, such as proliferation and differentiation [198,199,200]. However, while there is compelling evidence of ERK1/2 involvement in different cell death modalities, ERK1/2 ambivalent and controversial effects made its understanding in cell death daunting.

Necroptosis, usually defined as a caspase activity-independent RIPK1/RIPK3/MLKL-mediated cell death program [201], promotes the gene expression of proinflammatory cytokines and immunogenic molecules release upon the characteristic plasma membrane rupture [202]. Preclinical studies have revealed an important role of necroptosis in various disease processes, and inhibition of necroptosis in mouse models has been shown to be beneficial [203]. Nevertheless, clinical trials proposing necroptosis inhibitors in patients are still in their early days, and only a few inhibitors with appropriate in vivo properties have been developed [204]. Since the induction of necroptosis has profound differences in the outcome of a pathological situation [205,206], ongoing research is geared towards discovering novel inhibitors or identifying new regulatory molecules. In the DDT, the researchers are interested in identifying necroptosis modulators that might short-circuit the canonical signaling pathway and for which FDA-approved compounds exist, thus allowing for drug repurposing strategies.

The team’s recent findings, and that of others, show ERK1/2 involvement in necroptosis-activated-cell-autonomous functions via the increased expression of proinflammatory cytokines genes [205,206]. Focusing on quantitative analysis of ERK1/2 signaling dynamics using kinase activity reporter imaging, the team revealed distinct amplitude- and frequency-modulated (AM/FM) ERK1/2 activity signaling dynamics depending on the triggered cellular process: survival, apoptosis or necroptosis [207]. The DDT team’s results support the idea that the early onset of AM/FM ERK activity dynamics mediates the proinflammatory cytokine gene expression increase during TNF-induced necroptosis in L929 cells. To follow up on this innovative concept, they are currently investigating whether the rewiring of ERK1/2 signaling dynamics could modulate the immunogenic consequences of necroptosis and identify conditions that contribute to this process.

### 2.24. Team Suzanne

The team of Dr. Magali Suzanne at the CBI in Toulouse had an initial main interest in understanding how apoptosis contributes to morphogenesis.

The contribution of apoptosis to morphogenesis was initially envisioned as participating passively to tissue remodeling [208]. Indeed, apoptosis was mainly considered as a process counterbalancing cell division to regulate cell number. Further studies suggested that apoptosis can impact the surrounding tissue, although the cellular mechanism was unclear [209,210,211]. A pioneer study from the Suzanne team revealed that apoptotic cells, far from being eliminated passively, are “force-generating” cells and actively trigger remodeling of the surrounding tissue [212]. To do so, the apoptotic force relies (1) on a dynamic reorganization of the actomyosin cytoskeleton at the onset of apoptosis, which forms an apico-basal cable-like contractile structure and (2) on the maintenance of cell–cell adhesion, which allows the transmission of this force to the surrounding cells. The team further investigated which cellular mechanism is responsible for force generation and found that the nucleus constitutes an essential anchoring point for the apico-basal actomyosin cable in order to produce an apoptotic force [213]. This original cellular mechanism appears to be conserved in vertebrates as shown by the implication of apoptotic force in neural tube bending in avian embryos. In this model, the deformation generated by the apoptotic cells on their neighbors persists longer as the dorsal neural tube bending goes on, suggesting a rachet-like mechanism leading to the progressive curvature of the tissue [214]. Altogether, these works demonstrated that apoptosis has a mechanical impact on the surrounding tissue, a property that is conserved during evolution. These data brought apoptosis to the field of biomechanics, and complement recent findings showing that apoptotic cells also respond to mechanical signals coming from their microenvironment [115,215]. The team is currently testing whether this mechanical role of apoptotic cells has an impact on tumor development.

Finally, the team showed that an apico-basal force, similar to the one generated in apoptotic cells, is also generated by cells undergoing EMT [216]. The team is currently investigating how this EMT force compares to the apoptotic one. Is this capacity to generate force before being extruded a common process shared by both types of cells independently of the outcome (life or death)? Could this process be regulated by non-apoptotic functions of caspases or are they totally different mechanisms regulated by different signaling pathways? These are the questions currently being addressed by the Suzanne team.

### 2.25. Team Susin

Dr. Santos A. Susin and Prof. Florence Nguyen-Khac co-lead the team “Drug resistance in hematological malignancies (DRIHM)” at the Centre de Recherche des Cordeliers (CRC) in Paris. The main objective of the team is to analyze the molecular mechanisms associated with drug resistance (apoptotic avoidance) in different hematological malignancies, especially chronic lymphocytic leukemia (CLL). To develop this program, Dr. Susin and Prof. Nguyen-Khac assembled a team composed of researchers that specialize in cell death and drug resistance and clinicians working in the pathophysiology of lymphoproliferative disorders.

CLL is an incurable disease with a heterogeneous clinical course and becomes frequently resistant, even to the newly developed targeted therapies [217]. This leukemia is characterized by an accumulation of monoclonal CD5^+^ B cells in the peripheral blood, bone marrow and secondary lymphoid organs. CLL prognosis is dependent on clinical staging and biological markers, including *IGHV* status and molecular and chromosomal abnormalities. Among them, one of the main projects of our team focuses on the assessment of the gain of short arm of chromosome 2 (2p gain), a rare but recurrent cytogenetic abnormality in CLL, linked with poor prognostic factors and apoptotic avoidance [218]. The team initially demonstrated that 2p gain was associated with overexpression of *XPO1, TTC27, BCL11A, REL, AHSA2* and *USP34*, and that *XPO1* plays a central role in drug resistance [219]. They are now unraveling the specific role of these genes to determine whether they cooperate with *XPO1* in the apoptotic avoidance characterizing 2p gain. In a more translational perspective, the team is also interested in proposing original approaches to overcome the therapeutic relapse characterizing CLL. Of note, the current chemotherapeutic treatments induce cytotoxicity via a caspase-dependent mechanism with a rather variable outcome. Indeed, the CLL B cells present molecular defects that make them particularly resistant to the caspase-dependent pathway (TP53 inactivation, overexpression of MCL1 or BCL2). Therefore, the introduction of new drugs inducing cell death via alternative caspase-independent mechanisms could provide new means of improving the current strategies against CLL. To this end, the DRIHM team has recently demonstrated that the ligation of the CD47 receptor by agonist peptides induced caspase-independent cell death in CLL cells, with no detected resistance. Importantly enough, CD47 ligation induces cell death rapidly, with a higher efficacy in CLL cells, and reduced tumor burden in a CLL mouse model [220,221,222,223]. Therefore, the use of CD47 agonist peptides emerges as a potential chemotherapeutic against CLL. The DRIHM team is currently evolving this new therapeutic approach and exploring new opportunities to overcome CLL apoptotic blockade, such as the targeting of the CLL metabolism or the implication of the leukemic microenvironment in CLL drug resistance, with a particular focus on the role of the extracellular vesicles, a messenger between circulating CLL tumor cells and the stromal cells in the lymph nodes [224].

## 3. Conclusions

Instead of reaching a plateau, research into regulated cell death has dramatically thrived and diversified in the last few years. The underlying major causes for this are:

(1) Understanding that many proteins originally associated with apoptosis also have non-apoptotic functions, some of them being vital both in normal and cancer cells. The characterization of the mechanisms responsible for the control of these non-apoptotic functions is among the grand challenges in the coming years.

(2) The characterization of novel canonical and non-canonical mechanisms to induce apoptosis in cancer cells resistant to apoptosis. This is particularly important for developing efficient anti-cancer therapies.

(3) A better mechanistic comprehension of regulated forms of necrosis, which are currently considered as an emerging strategy for eliminating cancer cells resistant to apoptosis. In particular, deciphering the anti-cancer immune response induced by the different forms of regulated necrosis will open new therapeutic avenues.

(4) The growing evidence that most PCD modalities are involved in various pathologies, such as neurodegenerative diseases and cancer. Understanding this causality from a mechanistic perspective will help identify and target key regulators to halt the progression of these crippling diseases.

In conclusion, the FCDRN will continue to federate research on cell death by providing the framework for scientific interactions and organizing national and international meetings in the future.

## Figures and Tables

**Figure 1 biomolecules-12-00901-f001:**
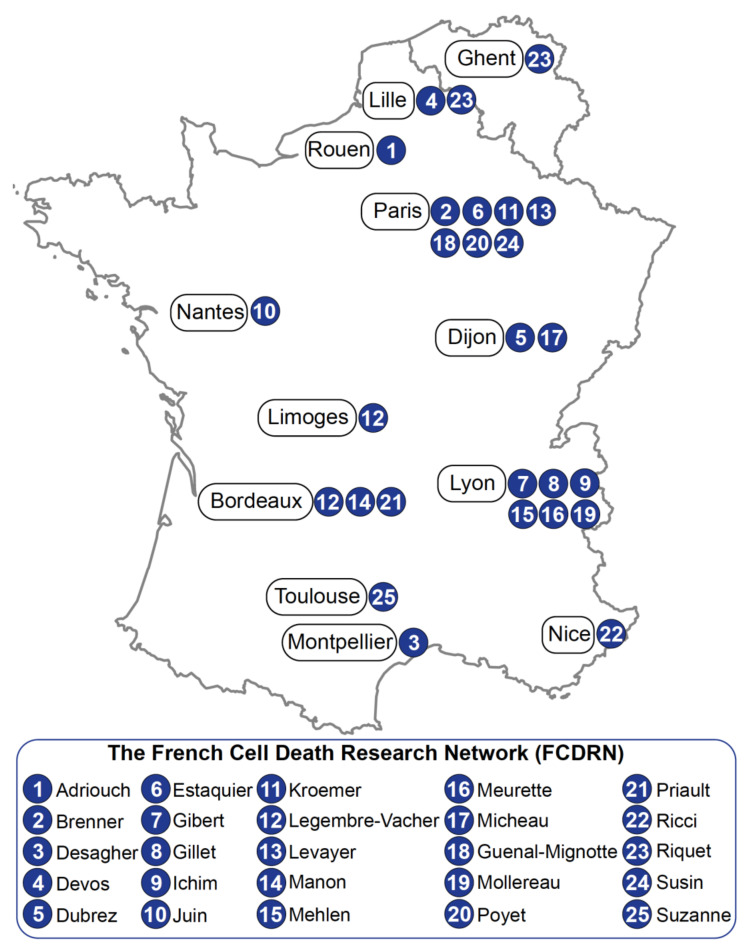
Geographical distribution of the FCDRN research teams.

## Data Availability

Not applicable.

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
