# Peer review of "Keeping Cell Death Alive: An Introduction into the French Cell Death Research Network"

_biomolecules, 2022, doi:10.3390/biom12070901_

Round 1

Reviewer 1 Report

This review has attempted to illuminate the current state on cell death research in the French Cell Death Research Network (FCDRN) community. A convenient description of the main research centers in France that have made significant contributions to cell death research is given.

- The review is well structured, but it is worth noting one general observation: not all of the groups include the current topics of work (team Legembre-Vacher and others). I recommend to include this information to the appropriate sections.

- In the sentence “Ferroptosis has been established as a novel form of regulated necrosis that is clearly distinct from other known cell death pathways.” (P.7 L. 304 ) please, add a reference.

- There are a uniform stylistic errors in the text - the transition from the form "they, them" to "we, ours".

Examples of the error:

P. 6 L. 239 - we showed and H202 (as a necrosis inducer)…..

P.  6 L. 248 - we observed that increasing ATP production favors ….

P. 12 L. 559 -  we wonder whether …..

L. 563 - we would like to address is whether the CD95…..

P. 13 L. 578 - For instance, we still know very……

P. 14 L. 629 - Furthermore, we contributed to the investigation…

L. 631 - We are now interested in defining the role …..

L. 634 - we will get a better knowledge….

P. 17 L. 785 - Our working hypothesis is ….

Paragraph 801-808 - Our team …. We have demonstrated that…… We have clarified the regulation….Finally, we are deciphering…. Р. 18 L. 877 - In the DDT, we are interested in ……

L. 882 - Our recent findings, and that of others

Р. 19 L. 886 - Our results support the idea that the ….

Minor blunders:

P. 13 L.616.– objects (isolated mitochondria...) or artificial objects (liposomes...). --Why is a multiple dots left out?

P. 14 L.  661 Duplication of the term addition:  “In addition, additional basic and translational research …”

In any case, these are minor points and the review is ready for publication.

Author Response

We would like to thank the reviewer for his comments. We have made the minor corrections suggested in the revised version of the manuscript.

Reviewer 2 Report

This manuscript is an overview of members of the French cell death research network (FCDRN), that includes 25 research labs located in France, and Belgium. The research network undertakes multiple approaches for delineating the regulatory mechanisms of regulated cell death, such as evolutionary approaches in work carried out in the yeast and fly model organisms, examining the non-apoptotic roles of key regulators of apoptotic cell death and exploring the roles of the novel forms of regulated cell death in the pathogenesis of several diseases and in novel therapeutic strategies. The work of the research network covers a wide range of cell types and animal models, cell death modalities as well as basic cell biology, infection immunology and cancer biology.

In the paper, a brief summary of each contributing lab is included, with details of the research focus and a brief summary of the data driving the hypotheses is given.

In general, the text is well written and the essential essence of each research groups work is easy to extract from each section. It is interesting to see in which general direction the different labs are taking to drive our understanding of programmed cell death in both health and disease in France.

Author Response

We would like to thank the reviewer for his positive comments.

Reviewer 3 Report

In this manuscript the authors describe ongoing efforts in cell death research throughout France and Belgium, and define the French Cell Death Research Network (FCDRN) as the combined efforts of over two dozen groups geographically dispersed throughout the region. The authors make the point that, despite predictions that the cell death field would plateau with time, the field has actually ventured into new questions and endeavors and remains very much vibrant. The manuscript highlights the work of the FDCRN investigators, and the reader gets a strong sense that these researchers will continue to move the field forward. However, the manuscript comes across as a simple cataloguing of individual research efforts, but does not significantly highlight the network aspect of the FCDRN. Discussion of communication and synergy among group members, and a statement of the purpose or mission of the network, would convince readers that FCDRN is truly a network and not merely a spreadsheet of investigator names and research interests.

Author Response

We would like to thank the reviewer for his comment. This is a good point to emphasize the role of FCDRN in fostering scientific exchanges. In response to his comment we have added sentences in the introduction and conclusion to stress this point as followed:

On page 4:

The French Cell Death Research Network (FCDRN) is actively taking part in further understanding apoptosis in development, physiopathology and in the emergence of new research fields related to cell death. During the last few years, the FCDRN has organized several meetings between its members to foster discussion and collaborations. During these meetings, FCDRN researchers felt the need to spread the idea that the cell death field has dramatically evolved, expanded and is still vivid. In this review, we compile a non-exhaustive list of 25 contributions on cell death research from both France and Belgium (Figure 1). It is an attempt to reflect the current state on cell death research in the FCDRN community, and can be summarized as follows:

On page 29

In conclusion, the FCDRN will continue to federate research on cell death by providing the framework for scientific interactions, organizing national and international meetings in the future.